# Targeting Aging Pathways in Chronic Obstructive Pulmonary Disease

**DOI:** 10.3390/ijms21186924

**Published:** 2020-09-21

**Authors:** Molly Easter, Seth Bollenbecker, Jarrod W. Barnes, Stefanie Krick

**Affiliations:** 1Division of Pulmonary, Allergy and Critical Care Medicine, Department of Medicine, The University of Alabama at Birmingham, Birmingham, AL 35294, USA; measter@uab.edu (M.E.); setheb@uab.edu (S.B.); jbarnes@uabmc.edu (J.W.B.); 2Gregory Fleming James Cystic Fibrosis Center, The University of Alabama at Birmingham, Birmingham, AL 35294, USA

**Keywords:** COPD, aging, senescence, inflammation, senolytics, cigarette smoke

## Abstract

Chronic obstructive pulmonary disease (COPD) has become a global epidemic and is the third leading cause of death worldwide. COPD is characterized by chronic airway inflammation, loss of alveolar-capillary units, and progressive decline in lung function. Major risk factors for COPD are cigarette smoking and aging. COPD-associated pathomechanisms include multiple aging pathways such as telomere attrition, epigenetic alterations, altered nutrient sensing, mitochondrial dysfunction, cell senescence, stem cell exhaustion and chronic inflammation. In this review, we will highlight the current literature that focuses on the role of age and aging-associated signaling pathways as well as their impact on current treatment strategies in the pathogenesis of COPD. Furthermore, we will discuss established and experimental COPD treatments including senolytic and anti-aging therapies and their potential use as novel treatment strategies in COPD.

## 1. Introduction

COPD is a progressive and debilitating respiratory condition that affects millions of people worldwide and represents a large medical and financial burden [1]. The worldwide population is living longer by 2050, 21% of the world population will be 60 years or older [2] and the prevalence of age-related chronic diseases like COPD is also increasing along with morbidity, mortality, and a greater degree of disability [3,4,5,6]. There is a five-fold increased risk for those aged over 65 years compared with patients aged less than 40 years of developing COPD [7]. In a recent report analyzing a cohort of 126,283 COPD patients (≥45 years), the prevalence of COPD was 3.6% and it increased greatly with age (1.9% in 45–64 years, 4.8% in 65–74 years, 6.8% in 75–84 years and 5.6% in ≥ 85 years) [8]. In addition, the elderly showing age-associated changes of the lung [9], which are per se not a pathological but it is unclear, whether those alterations might predispose to COPD development.

The Global Initiative of Obstructive Lung Disease (GOLD) has proposed guidelines for management and treatment of COPD; however, therapeutic interventions targeting the elderly population still need to be evaluated and translated into clinical practice [10,11]. COPD is considered a disease of accelerated aging with several aging pathways involved in its pathogenesis [12,13,14]. Current therapies mainly focus on symptomatic treatment. Thus, the understanding of age-related pathomechanisms associated with COPD is critical to uncover novel disease specific treatment strategies.

### Impact and Pathophysiology of COPD in the Elderly

COPD has become the third leading cause of death worldwide with the incidence increasing with age [15]. It has been reported that COPD is 2–3 times more prevalent in the elderly, and given that the world population is aging, age will become an even more significant risk factor for COPD [16].

COPD is a chronic inflammatory disease, and it is known that increased inflammation results in a progressive decline in lung function over time [17,18]. Its pathophysiology includes irreversible airway limitation caused by long term inhalation of noxious gases and/or cigarette smoke [17,19]. Classically, COPD has been defined by two main pathologies: (1) emphysema with parenchymal destruction and loss of alveolar septa; and (2) chronic bronchitis that consists of chronic bronchial inflammation of the bronchi, which is clinically defined as a chronic cough and sputum production for 3 months for two continuous years [20]. The diagnosis is usually based on symptoms and obstructive lung function test (spirometry) values with patients often having various degrees of severity of chronic bronchitis, emphysema, or an overlap of both [11].

Particularly elderly patients with COPD suffer from additional chronic diseases such as coronary artery disease, thyroid dysfunction, hypertension, congestive heart failure, atrial fibrillation, and other cardiovascular diseases [21]. Individuals with COPD have a four-fold increase risk for cardiovascular disease, and a significantly increased risk of psychiatric, renal, neurological, gastrointestinal, and metabolic comorbidities [22,23]. These aforementioned COPD-associated comorbidities have been linked to aging-associated pathomechanisms including chronic inflammation, cell senescence and telomere shortening [23,24,25,26,27,28,29]. In addition to these mechanisms, dysregulation of the extracellular matrix including protease/anti-protease balance and dysregulated matrix metalloproteinases [30,31] also play a role in the pathogenesis of COPD in the elderly [13]. Those comorbidities and their associated pharmacological treatments make COPD therapies in the elderly even more complex and challenging.

## 2. Aging-Associated Molecular Mechanisms in the Pathogenesis of COPD

Aging is a progressive degeneration of tissues that has negative effects on structure and function of vital organs [32]. Age-related diseases develop when physiological anti-inflammatory and antioxidant mechanisms fail to protect the body from damage due to a persistent low-grade inflammation and increased reactive oxygen species (ROS). This imbalance leads to cell and/or tissue injury, which contributes to COPD in the elderly in combination with the general risk factors for COPD such as cigarette smoking.

Additionally, there is an age-associated decline in lung function and structural dysfunction starting in the third decade of life [33]. For example, the lung function of normal aging individuals resembles an accelerated aging profile of lung function decline (e.g., mild airflow limitation severity) in COPD consistent with early stages of the disease [12]. This loss of function in the elderly can be contributed to an underlying phenomenon that has been called the “senile lung”, which is characterized by alveolar enlargement in the absence of wall destruction [12,34,35]. In addition to lung function decline and these structural changes with age, the pulmonary defense mechanisms are less effective, leading to an increased risk of lung infections and a decreased protective response against oxidative stress and inflammation [36]. The senile lung, however, is pathophysiologically different from emphysema in COPD [9]. Several mouse models have been generated in an attempt to recapitulate the senile lung, including knockout mice deficient in klotho, the senescence marker protein-30 (SMP30), or the “Senescence-Accelerated Mouse” protein (SAM) [12,37,38,39]. In addition, Schulte and colleagues found age-related structural and functional changes in an aged mouse lung [39]. COPD animal models are a valuable tool for understanding the underlying molecular mechanisms due to the potential of genetic as well as pharmacological pathway modulation [40]. In conclusion, these mouse models demonstrate the age-associated molecular mechanisms involved in lung aging and their potential contribution to COPD independently of known mechanisms such as cigarette smoking.

### 2.1. Inflammaging

Aging-associated chronic low-grade inflammation (inflammaging) plays a major role in the pathogenesis of COPD and stems from an imbalance of inflammatory and anti-inflammatory networks [41]. This discrepancy increases with age, leading to increased disease susceptibility [41]. COPD-associated inflammaging is characterized by an increase in immune cells (e.g., alveolar macrophages, neutrophils, and T lymphocytes) that secrete cytokines, chemokines, growth factors and lipid mediators, which perpetuate the inflammatory process and destruction of the lung parenchyma [42]. In particular, interleukin-6 (IL-6) has been characterized as an age-related pro-inflammatory cytokine and is associated with negative health outcomes and mortality, and linked to persistent, low grade activation of chronic inflammation [43,44]. IL-6 also plays a major role in sepsis with IL-6 levels in serum correlating with onset and progression of the disease [45,46]. In addition, IL-6 was found to be increased in COPD patients with acute exacerbations [47] and was associated with the development of emphysema in COPD [48]. COPD lung cells exhibit a “senescence-associated secretory phenotype” (SASP), which is characterized by senescent lung cells secreting pro-inflammatory molecules such as IL-6, IL-8, monocyte chemoattractant protein-1, and plasminogen-activated inhibitor-1 [49,50], which further contribute to the pathogenesis of COPD.

### 2.2. Cell Senescence

Cell senescence is defined as an irreversible growth arrest in response to stressors such as telomere shortening, oxidative stress, stem cell depletion and persistent DNA damage [12]. Cell senescence was originally discovered in the 1960s by Hayflick when he observed that primary culture cells lost replicative capacity after several passages [51]. More recently, cellular senescence has shown to contribute to the loss of tissue homeostasis leading to the accumulation of senescent cells and contributing to the aging process [52,53,54]. Cell senescence has also been shown to contribute to the accelerated aging process in COPD [55,56]. Cigarette smoke can induce expression of the cell senescence marker p21 in epithelial cells and fibroblasts. Emphysematous lungs show increased expression of p16, p19 and p21, all of which are cyclin kinase inhibitors and markers of cell senescence [56]. Furthermore, senescent cells lose regenerative capacity and prevent cell repair in the COPD lung contributing to the progression of emphysema and worsening lung function over time.

### 2.3. Stem Cell Exhaustion

Stem cells play an important role in replacing cells lost to homeostatic turnover, injury, and disease [57]. With age, stem cell function and regenerative capacity decrease. Age-related decline in stem cell capacity in tissues that have regenerative potential can contribute to the development of age-associated diseases [57,58]. The lung is considered a slow renewing organ with a stable cell pool of differentiated cells and infrequent proliferation of stem cells [59]. Therefore, extensive lung injury results in limited proliferative capacity of stem cells and its consecutive depletion [60]. For example, airway basal progenitor cells play an important role in the lung’s response to injury, as they are multipotent cells that can differentiate into basal, secretory, and ciliated cells [61]. In COPD, the differentiation ability of airway basal cells is decreased, which is accompanied by an early exhaustion of airway basal progenitor cells [61,62]. Therefore, stem cell regenerative therapy could potentially benefit COPD patients, especially the elderly subpopulation.

### 2.4. Oxidative Stress

The free radical theory states that oxidative stress is a key driver of accelerated aging [63]. Oxidative stress arises from endogenous antioxidant defense mechanisms being overwhelmed by the presence of ROS [64]. This imbalance between pro-oxidants and anti-oxidants increases with age and leads to an accumulation of oxidative damage contributing to senescence-associated tissue injury [63]. Several studies have demonstrated that a reduction in oxidative stress leads to a longer lifespan in *C. elegans*, drosophila, and mice [63,65,66,67,68,69,70]. Furthermore, cigarette smoke exposure and mitochondrial dysfunction have been shown to lead to oxidative stress in COPD [71]. Oxidative stress causes DNA damage to lung cells thereby inducing additional aging hallmarks such as cellular senescence and inflammaging [72]. In summary, oxidative stress plays a role in age-associated pathomechanisms of COPD such as cellular senescence and inflammaging. Therefore, antioxidant treatments might be effective in targeting the accelerated aging process in COPD.

## 3. Current Treatment Options for Individuals with COPD

There is no cure for COPD; however, there are a variety of therapies that alleviate symptoms, mitigate the risk of complications, and enhance the quality and quantity of life. Current standard therapies include smoking cessation, bronchodilator therapy, oxygen supplementation, pulmonary rehabilitation, and lung transplantion [73,74,75,76,77].

Smoking cessation, when done effectively, can result in long term quit rates of up to 25% [78]. Smoking cessation is achieved through pharmacotherapies combined with counseling [77]. Pharmacotherapies consist of nicotine replacement products, including nicotine gum, nasal spray, transdermal patches, tablets, and lozenges; however, use of these products alone has a low success rate [79,80]. Counseling in combination with pharmacological products like varenicline, bupropion, and nortriptyline have been shown to be effective therapies for smoking cessation [81].

Both short- and long-acting inhaled bronchodilators, including beta adrenergic receptor agonists, and muscarinic antagonists are mainstay therapies in the pharmacological treatment regimen of COPD patients [43]. They achieve bronchodilation through the following different mechanisms: (1) muscarinic antagonists bind to M3 receptors in airway smooth muscle cells to block acetylcholine-mediated bronchoconstriction [82]; and (2) B2 agonists stimulate B2-adrenergic receptors inducing smooth muscle relaxation [83,84]. There is growing evidence that combination treatment of these long-acting bronchodilators is more beneficial than monotherapies for improving health-related quality of life, lung function, and decreasing COPD exacerbation frequency [85]. Inhaled corticosteroids (ICSs) are commonly used to treat asthma, but are also used for COPD in combination with bronchodilators, leading to a decrease in COPD exacerbations but an increased risk of pneumonia in some cases [86]. Furthermore, triple combinations are now available to treat therapy-refractory COPD patients or those with frequent exacerbations [85,87].

COPD patients with hypoxemia show an increased survival rate with supplemental oxygen therapy [88]. A more comprehensive therapy for COPD is pulmonary rehabilitation, which is a patient-tailored therapy that consists of exercise training, education and behavioral interventions. It is designed to improve the overall physical and psychological condition of COPD patients [89]. In addition, lung transplantation can be a treatment option for patients with end-stage COPD or rapidly progressing disease. Lung transplantation does not prolong survival; however, it can improve functional capacity and overall health, but advanced age is normally an exclusion criteria [90]. Double lung versus single lung transplant has been shown to provide longer survival, especially in patients older than 60 years [91]. COPD patients with severe emphysema can be treated with lung reduction surgery or endobronchial valve or coil placement to remove or reduce the emphysematous tissue, thereby decreasing hyper-inflation and providing the space of more functional lung tissue. These procedures have been shown to be beneficial for certain subgroups of COPD patients [92,93,94,95,96,97].

Together these therapies are effective in treating the symptoms of COPD, but they have not been shown to attenuate disease progression, and it is not clear whether they are as effective in the elderly population.

## 4. Anti-Aging Treatment Strategies in COPD

### 4.1. Repurposed Drugs as Potential Anti-Aging Therapies

#### 4.1.1. Resveratrol

Resveratrol (3,4,5-trihydroxystilbene) is a plant polyphenol that is part of a family of antibiotic compounds called phytolaxins that are produced by plants in response to infections [98]. Resveratrol can be found in many common foods such as grapes, red wine, and peanuts [98]. Previous studies have shown that resveratrol can extend the life span of certain organism including yeast, drosophila, *C. elegans*, and fish [99,100,101]. Resveratrol has anti-inflammatory and antioxidant properties, which make it a potentially useful treatment for age-associated COPD [102,103]. Specifically, in COPD patients, resveratrol has been shown to inhibit the release of IL-8 in alveolar macrophages by 94% and 88% in smokers and patients with COPD respectively [18]. Another study showed that treatment with resveratrol reduced C-reactive protein levels and increased total antioxidant status values in smokers [102]. In a COPD rat model, resveratrol demonstrated anti-inflammatory and anti-oxidant properties by decreasing IL-6 and IL-8 production, upregulating Sirtuin1 and proliferator-activated receptor-γ coactivator-1α (PGC-1α), lowering levels of malondialdehyde and increasing superoxide dismutase activity [104]. Sirtuins are well known for their anti-aging properties [105,106,107]. Additionally, resveratrol has shown to have therapeutic effects in a COPD mouse model by decreasing the levels of the pro-inflammatory cytokines IL-17, IL-6, TNF-α, and TGF-β, attenuating fibrosis and mucus hypersecretion [108]. Together, these studies suggest that resveratrol has anti-inflammatory and anti-aging properties and could be a potential future treatment for COPD patients.

#### 4.1.2. Metformin

Metformin is a drug widely used for the treatment of type 2 diabetes and polycystic ovary syndrome [109]. Metformin belongs to the drug class of biguanides. It suppresses endogenous glucose production mainly in the liver and increases glucose uptake in the muscle [110]. Metformin has been characterized as a ‘geroprotector’, a class of molecules that are known to have anti-aging properties [105]. Anisimov and colleagues showed that treatment with metformin increased life span by 14% and delayed tumor progression in female SHR mice [109]. Metformin activates the 5′ AMP-activated protein kinase (AMPK), which has downstream effects on the Sirtuin-forkhead box (FOXO) axis and Sirtuin-mTOR signaling. These are two important pathways for cellular response to oxidative stress which leads to attenuation of airway inflammation and lung function improvement as well as extends life span [111]. COPD is often associated with type 2 diabetes [112]. Two studies examined the effects of metformin on COPD outcomes. Hitchings and colleagues conducted a randomized control trial to examine the effects of short-term treatment with metformin on the frequency of severe COPD exacerbations in COPD patients without type 2 diabetes [113]. There was no difference in CRP and blood glucose levels after hospitalization from a severe exacerbation. This study suggests that short-term treatment with metformin is not sufficient to improve COPD symptoms. The long-term effects of metformin were not examined, which would be interesting to study since COPD is a chronic illness. A study by Bishwakarma et al. observed that patients with COPD and diabetes treated with metformin were less sick, less likely requiring oxygen, and had fewer hospitalizations when compared to patients that did not receive metformin [114].

#### 4.1.3. Melatonin

Melatonin, *N*-acetyl-5-methoxytryptamine, is a hormone produced in the pineal gland that plays a key role in circadian rhythm control [115]. In addition, melatonin is a free radical scavenger for OH, O_2_^−^, and NO [116,117]. Melatonin supplementation has been shown to increase the lifespan of mice, rats, and fruit flies [117,118,119]. In patients with COPD, melatonin levels were shown to be decreased during exacerbations in combination with an increase in oxidative stress and decreases in antioxidant enzymes [120]. A randomized, double-blinded controlled trial focusing on the treatment of COPD patients with melatonin showed a decrease in oxidative stress and dyspnea after melatonin treatment and an increase in IL-8 levels in non-treated control COPD patients [121]. Additionally, melatonin treatment has been shown to reduce MUC5AC, a mucin involved in COPD-associated mucus hypersecretion [122] and IL-6 production [123]. Another report demonstrated that mice exposed to cigarette smoke and treated intraperitoneally with melatonin showed decreased total cell counts in bronchoalveolar lavage fluid (BALF), as well as decreased transforming growth factor (TGF)-β1 [124]. Together, these studies suggest that melatonin has anti-oxidative and anti-inflammatory properties and therefore might be useful in the treatment of COPD after validation in appropriate clinical trials.

### 4.2. The FGF23/Klotho Signaling Pathway

In 1997, Kuro-o et al. fortuitously discovered that deletion of the protein α-klotho (klotho) in a mouse model led to a premature aging phenotype including features such as skin atrophy, osteoporosis, soft tissue calcification, gonadal dysplasia, infertility, hypoglycemia, arteriosclerosis, and pulmonary emphysema [125]. Furthermore, these mice died within 8–12 weeks of age. On the other hand, overexpression of klotho in mice extended their lifespan by approximately 30% [126,127,128]. In humans, serum levels of klotho decrease with age [128] and klotho seems to play a protective role in age-related neurodegenerative disorders such as Alzheimer’s Disease [129]. Klotho is a co-receptor for fibroblast growth factor (FGF) 23, a pro-inflammatory and phosphaturic hormone, which signals through two main receptors: FGF receptor 1 (klotho dependent signaling) and FGF receptor 4 (klotho independent signaling) [130,131,132]. FGF23 is secreted by osteocytes and is important for regulating phosphate [131]. The FGF23/klotho signaling pathway has been studied in several inflammatory diseases such as heart disease, chronic kidney disease (CKD), and inflammatory airway diseases including COPD [132,133,134,135,136]. FGF23 levels have also been shown to be higher in COPD patients with more frequent exacerbations [137]. In COPD, where FGF23 levels have been shown to be elevated, klotho levels were found to be reduced. These results align with the finding that airway inflammation is modulated via the FGFR4/NFAT signaling pathway [133].

Klotho has been shown to exert anti-oxidative, anti-inflammatory, and anti-proliferative functions in the heart, lung, and kidney [135,138,139]. In the lung, klotho was shown to inhibit bronchial IL-8 secretion via attenuation of TGF-β signaling. Furthermore, klotho deficiency in the lung led to impaired mucociliary clearance [135,138]. Thus, klotho supplementation could be a useful treatment for chronic airway inflammation associated with COPD. In addition, FGFR blocking antibodies could become a novel treatment strategy to target airway inflammation. There are several FGFR inhibitors in phase 2 clinical trials for hepatocellular carcinoma [140]. Taken together, FGF23/klotho signaling may have the potential to be an amenable target pathway in the development of anti-aging and anti-inflammatory treatments in COPD.

### 4.3. IL-6 as an Anti-Inflammatory Therapy in COPD

Since IL-6 plays a key role in the pathogenesis of COPD, novel therapies targeting the inhibition of IL-6 would be reasonable. Previous clinical trials have used IL-6 receptor blockers in diseases such as rheumatoid arthritis and Crohn’s disease [43,141]. Furthermore, current clinical trials are investigating the use of IL-6 receptor blockers for the treatment of diseases associated with chronic inflammation such as chronic kidney disease, rheumatoid arthritis, polycystic ovarian syndrome, and Crohn’s disease. In particular, clinical trials using the IL-6 receptor blocker tocilizumab for rheumatoid arthritis were effective in decreasing circulating IL-6 levels and improving symptoms and disease activity scores [43]. To date, IL-6 receptor blockers have not been explored in COPD, but they represent a potential novel direction, requiring detailed studies to define correct timing, for an anti-aging/anti-inflammatory COPD phenotype.

A subset of COPD patients present with a predominantly IL-5 mediated eosinophilic inflammation, which has shown to contribute to exacerbation frequency and lung function decline [142,143,144]. IL-5 mediated eosinophilic inflammation plays a critical role in asthma [142,145,146] and treatments with mepolizumab [147] and reslizumab [148], which both target IL-5, are approved for asthma therapy. Age-associated differences in eosinophil activity is unclear [149,150], but for the subset of COPD patients with eosinophilic inflammation, IL-5 therapy represents a promising anti-inflammatory treatment and needs to be further explored as a potential COPD treatment strategy.

## 5. Potential Novel Anti-Aging Treatments for COPD

### 5.1. Senolytics

In recent years, senolytics have been explored as a possible treatment to “therapeutically remove” senescent cells. The ability to reduce senescent cells may result in decreased inflammation and increase the efficacy of stem and progenitor cells for tissue repair [49]. It has been shown that elimination of senescent cells might prevent or delay age-related functional impairments and extend life span [151]. Baker et al. demonstrated the ability to specifically target senescent cells in a mouse model of inducible elimination of p16-positive cells (p16Ink4a- apoptosis through targeted activation of caspase) using AP20187, which was shown to augment apoptosis [152]. Elimination of p16-positive cells has been shown to delay age-related pathologies such as sarcopenia, cataracts, and loss of adipose tissue [49,152]. Moreover, Zhu et al. discovered that senescent cells had an increased expression of pro-survival networks that confer apoptotic resistance in senescent cells [153]. Using siRNA to silence expression of key players in this pathway including ephrins (EFNB1 or 3), PI3Kd, p21, BCL-xL, and plasminogen-activated inhibitor-2, eradication of senescent cells was successfully achieved. In addition, drugs like Dasatinib and Quercetin were characterized as senolytic agents. Treatment with Dasatinib and Quercetin reduced senescent cells in aged mice and improved cardiac function and carotid vascular reactivity after single treatment [154] Recently, clinical trials examined Dasatinib and Quercetin as senolytic treatment strategies for idiopathic pulmonary fibrosis (IPF) and diabetic kidney disease [155]. Results for the IPF clinical trial showed that treatment was feasible and senolytics alleviated physical dysfunction in IPF [156]. In these studies, Dasatinib and Quercetin reduced senescent cells in the skin and SASP proteins in the blood (IL-1, IL-6, and MMP-9) [155,157]. Other senolytics such as Navitoclax, which targets BCL-2, BCL-X(l) and BCL-w, was also demonstrated to be effective in removing senescent cells, but was associated with high toxicity causing blood cell dyscrasias [155]. Moreover, cardiac glycosides such as Ouabain and Digoxin exhibit senolytic effects by inhibiting BCL-2 family proteins via inhibition of Na+/K+ ATPase [155,158]. They also exhibit selective senolytic effects on p16-expressing human airway epithelial cells [155,158]. However, to date, there are no current clinical trials assessing senolytic therapies for COPD, which may be a valid future option to investigate. Collectively, these findings demonstrate that senescent cells can be targeted and eliminated in preclinical models and patients with age-related diseases (see Table 1). Targeting senescent cells may be a potential therapy for aging-associated diseases such as COPD.

### 5.2. Stem Cell Therapies

In recent years, mesenchymal stem cells (MSCs) have been studied as a potential treatment for COPD [159]. MSCs are multi-potent stem cells that have fibroblast-like morphology and the capacity to differentiate into chondrocytes, osteoblasts, adipocytes, and muscle cells under different environmental conditions [160]. In addition to their regenerative properties, MSCs have distinctive anti-inflammatory properties that make them a potential treatment option for a variety of inflammatory disorders including COPD [160]. MSCs secrete anti-inflammatory molecules such as IL-10 and modulate the release of the pro-inflammatory cytokines IL-1β, TNF-α, IL-6, and prostaglandin (PG)E2 [160,161,162]. MSCs release growth factors important for tissue repair like VEGF, HGF, EGF, and TGF-β1 [160,163,164,165]. Another approach to employ stem cell therapy in COPD is through “ex vivo bioengineering” of functional lung tissue for implantation [166], which uses 3D matrices or scaffolds seeded with lung stem cells cultured to form functional lung tissue capable of gas-exchange [167]. This method has been shown to be feasible in rodents, pigs, and human lungs; however, it has yet to produce lung tissue with robust gas-exchange capacity [166,167,168]. Bioengineering of the lung has great potential as a therapy for progressive COPD due to its ability to restore damaged lung tissue and improve lung function.

### 5.3. Antioxidant Therapy

Antioxidant agents such as thiol molecules, especially glutathione and polyphenols, can reduce inflammatory gene expression and inhibit cigarette smoke-induced inflammatory responses [99]. Moreover, sputum from patients with stable moderate-to-severe COPD and control subjects showed increased glutathione disulfides and nitrosothiols [169]. Several clinical trials have been investigating the effects of antioxidants in different diseases (e.g., the potential of oral supplementation of glutathione in pediatric cystic fibrosis patients) and have shown a significant decrease in bowel inflammation scores following glutathione supplementation [170]. Unfortunately, lung function tests were not performed. A different clinical trial investigated inhaled glutathione as a treatment strategy for cystic fibrosis but found no clinically relevant improvements in lung function, exacerbation frequency, and patient-reported outcomes [171]. However, Lamson and colleagues published a case study using nebulized glutathione for treatment of acute respiratory crisis secondary to emphysema and found rapid resolution and improvement in respiratory function [172]. Moreover, the erdosteine thiol compound has muco-active, antioxidant and anti-inflammatory properties in addition to functioning as a modulator for bacterial adhesiveness [173]. The EQUALIFE study found that treatment with erdosteine led to a significant decrease in exacerbations and shortened hospital stay [174]. Erdosteine could prove to be an effective treatment for elderly COPD patients, especially since they are more susceptible to respiratory tract infections. In summary, more studies are needed to determine whether antioxidant therapies are useful to combat the aging process in COPD.

## 6. Conclusions

The lifespan of the world population is increasing, which will further enhance the prevalence of age-related diseases such as COPD. The increased lifespan and age-related diseases will ultimately put a burden on our healthcare systems and significantly affect the quality of life of these patients. Current treatments for COPD are mainly addressing COPD-associated symptoms, but alternative treatment strategies are needed to evaluate the underlying mechanisms that lead to the development of COPD, especially in the elderly (Figure 1). Therefore, it is of high interest to focus on anti-aging treatment strategies, since the numbers of the elderly with COPD will continue to rise. There are already promising therapeutic targets for aging-associated COPD pathomechanisms; however, more research needs to be conducted to better understand these mechanisms and help to develop safe and effective anti-aging therapies.

## Figures and Tables

**Figure 1 ijms-21-06924-f001:**
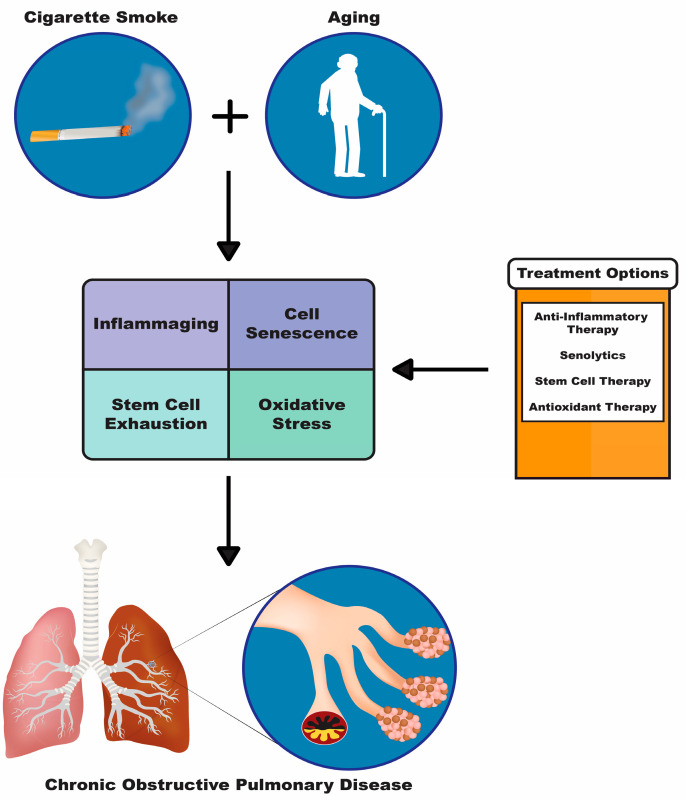
Diagram depicting two major risk factors for COPD cigarette smoke and aging. These risk factors cause downstream pathologies such as inflammaging, cell senescence, stem cell exhaustion and oxidative stress, which can be targeted with the treatment options shown in the orange and white box: anti-inflammatory therapy, senolytics, stem cell therapy and antioxidant therapy.

**Table 1 ijms-21-06924-t001:** A table of senolytic drugs. This table includes the molecular targets of each of the drugs and the progress to becoming a therapy for senescence.

Senolytics	Target	Model
Dasatinib	Tyrosine Kinases	Phase 2 Clincal Trial
Quercetin	Tyrosine Kinases	Phase 2 Clincal Trial
Navitovlax	BCL–2 Family	Animal Models
Cardiag Glycosides	BCL–2 Family	Animal Models

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
