# Peer review of "Targeting Aging Pathways in Chronic Obstructive Pulmonary Disease"

_ijms, 2020, doi:10.3390/ijms21186924_

Round 1
Reviewer 1 Report
Since the worldwide population 24 is living longer, the prevalence of age-related chronic diseases like COPD is also increasing along 25 with morbidity, mortality, and a greater degree of disability
I would suggest adding some numbers here : what is the prevalence among different age groups ?
It has been reported that COPD is 2-3 times more prevalent in the elderly and given 35 that the world population is aging, age will become an even more significant risk factor for COPD – again, I would use numbers here….
In particular, interleukin-6 (IL-6) has been characterized as an age-related pro-86 inflammatory cytokine and is associated with negative health outcomes, mortality, and linked to 87 persistent, low grade activation of chronic inflammation/
I would mention the major play of IL-6 in sepsis
Current therapies 132 include bronchodilator therapy, smoking cessation, oxygen supplementation, pulmonary 133 rehabilitation, and lung transplant [63-67].
I would start with " smoking cessation"
Author Response
Dear Reviewer 1,
Thank you very much for your constructive comments. We have included our answers here in a point-by-point response. Hopefully, our changes will make our manuscript acceptable for publication.
- Since the worldwide population 24 is living longer, the prevalence of age-related chronic diseases like COPD is also increasing along 25 with morbidity, mortality, and a greater degree of disability I would suggest adding some numbers here : what is the prevalence among different age groups ?
Answer: In the revised manuscript, we have included these numbers and their source in the revised introduction.
- It has been reported that COPD is 2-3 times more prevalent in the elderly and given 35 that the world population is aging, age will become an even more significant risk factor for COPD – again, I would use numbers here….
Answer: In the revised manuscript, we have included these numbers and their source in the revised introduction.
- In particular, interleukin-6 (IL-6) has been characterized as an age-related pro-inflammatory cytokine and is associated with negative health outcomes, mortality, and linked to persistent, low grade activation of chronic inflammation/ I would mention the major play of IL-6 in sepsis
Answer: Thank you very much for this suggestion. We have included this in the revised manuscript (lines 165-166).
- Current therapies 132 include bronchodilator therapy, smoking cessation, oxygen supplementation, pulmonary 133 rehabilitation, and lung transplant [63-67]. I would start with " smoking cessation"
Answer: Thank you very much for your suggestion. We have changed the order accordingly in the revised heading 3 (lines 222-251).
Reviewer 2 Report
This is a comprehensive and very well-structured review on the influence of age on current treatments for COPD.
In many places (Line 27, 63), the GOLD recommendations are cited but not referenced. Please provide the reference where indicated.
Line 65, I would suggest the Authors to refer to early stages of the disease rather than to GOLD stage I, which is no longer in use in clinical practice.
The term “senile emphysema” is largely used in this and other papers to refer to structural abnormalities that resemble those of emphysema. However, as the Authors properly state, there is no destruction of lung tissue in the aged lung. Therefore, I would strongly recommend to use the term “senile lung” whenever possible. In addition, I would like to see more discussion in the Introduction section on this topic, since this is crucial for the understanding of the review paper. Refer to: Bellia M. et al. Monaldi Arch Chest Dis. 2011.
The dysregulation of extracellular matrix has been proposed has an additional mechanism related to ageing that contributes to development of COPD. I would ask the Authors to comment on it.
The review covers the role of IL-6 as potential target. In this context, I wonder whether treatment specifically targeting IL-5 in a subgroup of COPD patients with high eosinophils should also be mentioned.
When addressing the role of antioxidants, the contribution of erdosteine (see the EQUALIFE study) should be included.
Table 1 shows the stage of development of the senolytic agents. Please specify at which phase of progress (for example: Phase 2, Phase 3a, …).
Author Response
Dear Reviewer 2,
Thank you very much for your constructive comments. We have included our answers here in a point-by-point response. Hopefully, our changes will make our manuscript acceptable for publication.
- This is a comprehensive and very well-structured review on the influence of age on current treatments for COPD.
Answer: Thank you very much.
- In many places (Line 27, 63), the GOLD recommendations are cited but not referenced. Please provide the reference where indicated.
Answer: We apologize for this omission. We have added the references in the revised manuscript.
- Line 65, I would suggest the Authors to refer to early stages of the disease rather than to GOLD stage I, which is no longer in use in clinical practice.
Answer: Thank you for this suggestion. We changed our revised manuscript accordingly (lines 94-95).
- The term “senile emphysema” is largely used in this and other papers to refer to structural abnormalities that resemble those of emphysema. However, as the Authors properly state, there is no destruction of lung tissue in the aged lung. Therefore, I would strongly recommend to use the term “senile lung” whenever possible. In addition, I would like to see more discussion in the Introduction section on this topic, since this is crucial for the understanding of the review paper. Refer to: Bellia M. et al. Monaldi Arch Chest Dis. 2011.
Answer: Thank you very much for pointing this out. We have changed our revised manuscript accordingly (see revised introduction and lines 91-103).
- The dysregulation of extracellular matrix has been proposed has an additional mechanism related to ageing that contributes to development of COPD. I would ask the Authors to comment on it.
Answer: We addressed this comment in the revised manuscript in lines 76-81.
- The review covers the role of IL-6 as potential target. In this context, I wonder whether treatment specifically targeting IL-5 in a subgroup of COPD patients with high eosinophils should also be mentioned.
Answer: Thank you for this suggestion. We have added those in the revised manuscript (lines 401-409).
- When addressing the role of antioxidants, the contribution of erdosteine (see the EQUALIFE study) should be included.
Answer: We have added this suggestion in the revised manuscript (lines 501-507).
- Table 1 shows the stage of development of the senolytic agents. Please specify at which phase of progress (for example: Phase 2, Phase 3a, …)
Answer: Thank you very much for this suggestions. We have edited the revised table accordingly.
Round 2
Reviewer 2 Report
No further comments, Happy with the revised version.